# Changes in Body Mass Index Among Korean Adolescents Before and After COVID-19: A Comparative Study of Annual and Regional Trends

**DOI:** 10.3390/ijerph22071136

**Published:** 2025-07-18

**Authors:** Seongjun Ha

**Affiliations:** Department of Physical Education, Korea National University of Education, Cheongju 28173, Republic of Korea; powerhah@hanmail.net

**Keywords:** body mass index (BMI), adolescents, COVID-19, socio-ecological model, health equity

## Abstract

This study aimed to longitudinally analyze changes in body mass index (BMI) among Korean middle and high school students before and after the COVID-19 pandemic. Data were obtained from the national-level Physical Activity Promotion System (PAPS), collected between 2018 and 2024. A total of 171,705 adolescents aged 13 to 18 were included in the analysis (86,542 males and 85,163 females), with a mean age of 15.2 years (SD = 1.68). Time-series analysis and two-way analysis of variance (ANOVA) were conducted to examine differences in BMI by year, sex, region (capital vs. non-capital), and urban–rural classification. The results indicated a significant increase in BMI during the pandemic period (2020–2022), peaking in 2022, followed by a gradual decline thereafter. Notably, male students and those living in rural or non-capital areas consistently exhibited higher BMI levels, suggesting structural disparities in access to physical activity opportunities and health resources. This study employed the Socio-Ecological Model and the Health Equity Framework as theoretical lenses to interpret BMI changes not merely as individual behavioral outcomes but as consequences shaped by environmental and policy-level determinants. The findings underscore the need for equity-based interventions in physical education and health policy to mitigate adolescent health inequalities during future public health crises.

## 1. Introduction

Adolescence is a critical period characterized by rapid physical, psychological, and social changes, during which individuals begin to form and consolidate long-term health behaviors and habits. [1] define adolescence as a distinct developmental phase between childhood and adulthood, spanning the ages of 10 to 24, and emphasize that experiences during this stage significantly shape lifelong health and behavioral outcomes. In this context, body mass index (BMI) is widely used as a physiological indicator to assess adolescent health status and predict the risk of chronic diseases in adulthood [2]. Due to its ease of measurement and high reliability, BMI has become a standardized metric for health assessment in school-based health screenings and national public health statistics [3].

Ref. [4] has recently identified adolescent overweight and obesity as pressing global public health issues. According to WHO, the prevalence of overweight and obesity among individuals aged 5–19 has more than tripled since 1975, a trend that is closely associated with societal changes such as urbanization, the expansion of technology-based learning environments, and increasingly sedentary lifestyles. Korea is no exception. National surveys conducted by the Ministry of Education and the Korea Disease Control and Prevention Agency indicate a steady increase in the average BMI of Korean middle and high school students over the past decade, along with stagnation or decline in physical activity participation rates [5].

These patterns raise serious socioeconomic concerns, as elevated adolescent BMI is linked not only to immediate health risks but also to long-term increases in the incidence of adult obesity, hypertension, diabetes, and cardiovascular diseases.

Against this backdrop, the global outbreak of coronavirus disease 2019 (COVID-19) brought about fundamental structural changes in the daily health behaviors and physical activity environments of adolescents. In Korea, nationwide school closures and remote learning policies were implemented to mitigate viral transmission, sharply reducing opportunities for everyday physical activity within school settings. Activities such as physical education classes, recess movement, after-school sports, and school clubs were suspended or restricted, and were instead replaced by online classes, screen-based learning, and prolonged sedentary behavior. As a result, physical activity levels decreased, screen time increased, and disruptions in sleep and dietary routines were widely observed [6,7]. These lifestyle shifts have raised public health concerns about increasing BMI levels and worsening health risks among adolescents.

A growing body of international research supports the notion that the pandemic has had structurally differentiated effects on adolescent health. For example, [8], through a systematic review based on the Socio-Ecological Model, emphasized that adolescent physical activity participation is determined by a multilayered set of factors, including individual, peer, familial, school, community, and policy-level influences. These environmental factors interact more dynamically in times of external shocks, such as the COVID-19 pandemic. Similarly, Ref. [9] proposed the Health Equity Framework, which explains that health disparities result from the interactions among power structures, social networks, individual conditions, and biological sensitivity, highlighting that adolescent health outcomes are rooted in structural inequities rather than merely individual choices. Furthermore, the findings of [10] suggest that socio-ecological, multilevel interventions can meaningfully reduce sedentary behavior in children, indicating that environmental interventions may significantly influence BMI trajectories among youth.

Importantly, the pandemic did not affect all adolescents equally. Differences in residential environments and educational conditions across gender, region, and urban–rural status likely amplified the disparities during the crisis. For example, students in non-capital or rural areas may have had limited access to physical education facilities or digital learning environments, and less access to public health resources and outdoor activity spaces. Gender-based differences in physical activity opportunities and attitudes toward exercise may also have affected responses to the pandemic [11,12,13].

Despite the importance of these issues, most existing domestic studies remain cross-sectional or short-term in nature, and very few have conducted time-series analyses to compare BMI changes before and after the pandemic. There is a noticeable research gap in longitudinal studies that examine adolescent BMI trends and regional or gender-based disparities in Korea. This limits the development of effective intervention strategies for schools and policymakers. Therefore, it is essential to conduct a systematic analysis of BMI trends using nationally representative longitudinal data, with particular attention to regional and gender equity.

Related international studies have reported similar findings. Ref. [14] found a significant association between reduced physical activity and increased BMI among children in Spain during the pandemic. Ref. [15] pointed out the limited effectiveness of digitalized physical education classes in maintaining student activity levels during school closures. In a large-scale study involving more than 200,000 adolescents from Latin America, Ref. [16] demonstrated that pre-existing gender disparities in physical activity were further exacerbated during the pandemic. While these studies highlight the negative impact of COVID-19 on youth physical activity and weight status, few have adopted a longitudinal and comparative design that visualizes intergroup differences over time.

Accordingly, this study utilizes national-level data from the Physical Activity Promotion System (PAPS), provided by the [17], to examine year-by-year changes in adolescents’ BMI, with particular focus on the impact of the COVID-19 pandemic. The analysis considers multiple intersecting variables, including region (capital vs. non-capital), urban–rural classification, and gender, to compare trends in BMI and identify disparities among subgroups. By doing so, the study aims to provide empirical evidence on the state of health inequality among Korean adolescents. Furthermore, it seeks to inform appropriate policy responses in school-based physical education and health systems during public health emergencies and to explore practical strategies for promoting adolescent health equity in the post-pandemic era.

## 2. Theoretical Background

To understand the factors influencing changes in BMI, it is necessary to move beyond simple obesity prevalence statistics and adopt a theoretical framework that systematically accounts for multiple, interrelated determinants—such as physical activity, educational environment, regional disparities, gender identity, and public policy. This study employs the Socio-Ecological Model (SEM) and the Health Equity Framework (HEF) to analyze the key structural factors and pathways shaping BMI changes within this multidimensional context.

### 2.1. BMI Determinants from a Socio-Ecological Perspective

The Socio-Ecological Model [18] posits that health behaviors are not solely the result of individual choices but are formed through interactive influences across multiple environmental levels, including individual, interpersonal, community, and policy systems. This model is particularly well suited for analyzing BMI, a health indicator that often reflects the cumulative impact of environmental complexity rather than a single causal factor. For instance, insufficient physical activity among adolescents may not stem from a lack of motivation alone but rather from a combination of structural constraints such as school curriculum design, the accessibility of local sports facilities, parental health awareness, peer influences, and the distribution of community sports infrastructure.

In a systematic review grounded in the SEM, Ref. [8] emphasized that adolescent physical activity is shaped not only by individual characteristics but also through the dynamic interactions of teachers, families, communities, and policy contexts. These multilevel environmental factors become even more influential during external shocks such as a global pandemic. In this study, BMI is thus interpreted as a health outcome embedded in an ecological context, shaped by the interplay of structural forces that either support or constrain physical activity opportunities. Ref. [10] further demonstrated that multilevel interventions based on the SEM are effective in reducing sedentary behavior in children, particularly when strategies encompass schools, families, communities, and policy systems. Their findings reinforce the legitimacy of using an ecological approach to understand and improve adolescent health behaviors.

### 2.2. Health Equity Perspective and Gender–Regional Intersectionality

Health equity theory posits that disparities in health outcomes are not merely the result of individual choices or behaviors, but stem from unequal distributions of social resources and structural barriers. This framework is especially relevant in the Korean context, where gaps in education and public health infrastructure exist between metropolitan and non-metropolitan regions. For example, adolescents in rural or non-capital areas often face significant disadvantages in terms of physical education hours, facility access, qualified instructors, and the availability of community health resources—all of which may affect physical activity participation and BMI changes [12].

Ref. [9], through the HEF, emphasize that health outcomes are shaped by avoidable and unjust structural disparities. They argue that realizing health equity requires a multidimensional approach that considers not only individual-level factors, but also relational networks, power structures, biological susceptibility, and institutional arrangements. This study adopts the perspective of the HEF to empirically examine whether disparities in BMI changes by region and gender during the COVID-19 pandemic—an exogenous public health crisis—can be attributed to underlying structural inequities.

Moreover, gender is treated not merely as a biological characteristic but as an intersectional factor that interacts with cultural expectations around body image, participation patterns in physical education, and psychological attitudes toward exercise. Ref. [16], analyzing adolescent data from Latin America, found that male students consistently engaged in higher levels of physical activity than females, while female students exhibited significantly longer sedentary time. These findings suggest that gender may condition how adolescents respond to constraints such as those imposed during the pandemic, thereby necessitating gender-responsive policy interventions.

### 2.3. Structural Constraints and the Effectiveness of School-Based Physical Education During the Pandemic

School-based physical education has traditionally served as a public mechanism to provide equitable access to physical activity for adolescents. However, as shown by [15], the shift to online instruction during the COVID-19 pandemic significantly undermined the effectiveness of school physical education, leading to a meaningful reduction in actual physical activity levels. This structural limitation disproportionately affected students with fewer opportunities for extracurricular physical activity, particularly those in low-income or rural settings. Given these circumstances, this study focuses on how pandemic-induced changes in school participation environments may have contributed to widening BMI disparities across social groups, and highlights the need to re-examine the protective functions of school-based health and physical activity systems.

#### The Analytical Focus of the Present Study

This study conceptualizes BMI as a key health outcome during adolescence and proposes an analytical framework built around three core dimensions:(1)Temporal changes—How did structural conditions before and after the COVID-19 pandemic shape BMI trends?(2)Regional disparities—How did differences in physical education and health infrastructure between metropolitan/non-metropolitan and urban/rural areas affect BMI changes?(3)Gender-based interactions—How did male and female students differ in their BMI trajectories and physical activity levels under pandemic constraints?

By integrating these dimensions, this research goes beyond describing obesity prevalence to empirically map the structural geography of adolescent health inequities.

## 3. Methods

### 3.1. Participants and Data Collection

This study was conducted using data from PAPS, which is implemented annually by the Korean Ministry of Education. PAPS is a nationwide physical fitness assessment system designed to systematically evaluate the health-related physical fitness of elementary, middle, and high school students, in accordance with the School Health Act Enforcement Decree and the Basic Plan for Student Health Promotion. The program is jointly administered by the Ministry of Education and regional Offices of Education, with measurements and data collection conducted by each school at least once a year.

Jointly managed by the Ministry of Education and metropolitan and provincial offices of education, PAPS assessments are conducted at least once a year at the school level. The program has been implemented nationwide since 2009 and typically takes place between April and October each academic year. Measurements are carried out under the supervision of physical education or health teachers, and the results are recorded in the National Education Information System (NEIS). These records are then aggregated at the municipal and national levels to inform Korea’s public health statistics.

The assessment components of PAPS focus on health-related physical fitness and include five areas: cardiorespiratory endurance, muscular strength and endurance, flexibility, power, and body composition. Among these, the primary indicator analyzed in this study, BMI, is automatically calculated by measuring each student’s height and weight and applying the internationally standardized formula: BMI = weight (kg) ÷ height^2^ (m^2^). The resulting BMI values are categorized according to age- and sex-specific reference standards provided by the Ministry of Education, classifying students into categories such as underweight, normal weight, overweight, and obese. This indicator is used not only for individual health guidance but also as a reference for school-level health education initiatives.

For the purposes of this study, PAPS data from six years (2018, 2019, 2020, 2022, 2023, and 2024) were analyzed. The year 2021 was excluded due to incomplete measurements stemming from school closures during the height of the COVID-19 pandemic. The final dataset included a total of 171,705 male and female students between the ages of 13 and 18, enrolled in grades 7 through 12 (i.e., middle and high school). The data were categorized by year, gender, region (capital vs. non-capital), and urban–rural classification (metropolitan city, regional city, province).

In terms of Korea’s administrative geography, the nation is divided into capital and non-capital regions. More specifically, there is one special city (Seoul), six metropolitan cities (e.g., Busan, Daegu), one special autonomous city (Sejong), eight provinces (e.g., Gyeonggi, Gangwon), and one special autonomous province (Jeju). Seoul, the capital, serves as the political, economic, and cultural center of the country. Metropolitan cities are large urban centers with populations exceeding one million and serve as regional industrial and transportation hubs. Provinces encompass both urban (“si”) and rural (“gun”) areas, making them spatially and socioeconomically diverse. The capital region comprises Seoul, Incheon, and Gyeonggi Province, and is home to over half of the national population. It is a key concentration zone for educational, economic, and cultural resources.

The dataset used in this study was provided by the Ministry of Education in a fully de-identified format. BMI values were extracted and classified by year, gender, and region, and subsequently cleaned for statistical analysis. Given the standardized nature of PAPS data collection across years and schools, the dataset is highly reliable for both longitudinal and cross-sectional analyses of adolescent health indicators in Korea.

Table 1 presents the general characteristics of the study participants. Table 2 presents the BMI classification standards used in PAPS, as provided by the Ministry of Education. These age- and sex-specific BMI thresholds are based on anthropometric data from Korean adolescents and recommendations from WHO and the Korean Society for the Study of Obesity (KSSO). The cutoff points between categories are known to have been established through a combination of statistical distribution and clinical guidelines.

### 3.2. Data Analysis

In this study, SPSS 26.0 was used to analyze changes in adolescents’ BMI across years, sex, regions, and urban–rural classifications. Descriptive statistics were first computed to determine the basic distributional characteristics of BMI, including means, standard deviations, and minimum and maximum values. This process was conducted separately by year, sex, region, and urban–rural classification, allowing for a visual and statistical understanding of overall BMI trends before and after the COVID-19 pandemic, as well as differences across subgroups.

To examine differences in mean BMI across years and urban–rural groups, one-way ANOVA was performed. In addition, independent samples t-tests were used to directly compare mean BMI between sex and region groups. These tests enabled a quantitative comparison of disparities by sex and region and provided theoretical support for the study’s discussion of health inequities.

To further investigate the interaction effects of year with other variables, such as gender, region, and urban–rural classification, two-way ANOVAs were conducted. Specifically, interaction terms were tested for year × gender, year × region (capital vs. non-capital), and year × urban–rural classification. These analyses provided insights into whether trends in BMI varied significantly across subgroups, or whether certain groups experienced sharper increases or decreases, particularly during the pandemic period between 2020 and 2022.

Finally, key findings were visualized using line graphs to illustrate time-series trends. Mean BMI trajectories were separately plotted by gender and region, enabling intuitive interpretation of changes across the pre- and post-pandemic periods. All statistical tests were conducted at a significance level of α = 0.05. To control for Type I error due to multiple comparisons, Bonferroni adjustments were applied where necessary.

### 3.3. Ethical Considerations

This study was conducted using data from PAPS, provided by the Ministry of Education of the Republic of Korea. All data used in the analysis were from fully anonymized secondary public datasets and contained no personally identifiable information. In accordance with Article 2, Paragraph 2 of the Enforcement Decree of the Bioethics and Safety Act of Korea, this study does not constitute “human subjects research” and is therefore exempt from review by an Institutional Review Board (IRB).

Moreover, the rights, safety, and privacy of the individuals represented in the dataset were thoroughly protected throughout the process of data acquisition and analysis. The researchers were not involved in any direct interaction with participants, nor in any stage of physical measurement or data entry. All data were used exclusively for research purposes and were never disclosed or transmitted to third parties without authorization.

This study also adhered to the ethical principles of the Declaration of Helsinki, established by the World Medical Association, which guides ethical conduct in research involving human participants. All stages of the study were conducted in compliance with academic research ethics and public data utilization guidelines. The research was designed and carried out in accordance with internationally accepted standards for ethical research involving human-related data.

## 4. Results

### 4.1. Analysis of Between-Group Differences in Mean BMI

As shown in Table 3, the results of the one-way analysis of variance (ANOVA) revealed statistically significant differences in mean BMI across the five measured years (excluding 2021) (F = 385.173, *p* < 0.001). This indicates that the average BMI of adolescents changed significantly over time, including both the pre-pandemic and post-pandemic periods. Notably, BMI peaked in 2022 and then slightly decreased in the subsequent years. These results suggest that disruptions to regular schooling—such as school closures and the transition to remote learning—may have contributed to reduced physical activity among students during the pandemic.

In addition, significant differences in BMI were observed according to urban–rural classification (F = 294.642, *p* < 0.001). Specifically, students residing in provincial (rural) areas had higher average BMIs compared to those in special or metropolitan cities. This disparity may reflect structural differences in access to physical education and healthcare infrastructure, as well as varying capacities for health-related guidance at the household level between rural and urban areas.

Table 4 presents the comparison of mean BMI by gender and region. Male students (M = 22.78, SD = 1.91) had significantly higher BMI scores than female students (M = 21.63, SD = 1.98), t(df) = 121.36, *p* < 0.001. Additionally, students residing in non-capital regions (M = 22.33, SD = 2.31) exhibited significantly higher mean BMI than those in the capital region (M = 22.02, SD = 1.47), t(df) = −30.614, *p* < 0.001.

These findings suggest that both gender and regional factors contribute to disparities in adolescent health outcomes. In particular, the higher BMI observed among students in non-capital areas may reflect an unequal distribution of physical education resources, differences in access to health services, and varying lifestyle environments, thereby underscoring the structural nature of health inequities between regions.

### 4.2. Interaction Effects Analysis

Table 5 presents the results of two-way ANOVA tests examining the interaction effects between year and key categorical variables. All interaction effects were found to be statistically significant. Specifically, significant interactions were observed between year and urban–rural classification (F = 6.687, *p* < 0.001), year and gender (F = 227.687, *p* < 0.001), and year and region (capital vs. non-capital) (F = 19.192, *p* < 0.001).

These findings indicate that changes in BMI over time did not occur uniformly across subgroups; rather, the patterns of change differed depending on gender, region, and urban–rural status. In other words, the impact of the COVID-19 pandemic and subsequent recovery periods manifested differently across population segments, thereby highlighting the importance of considering interaction effects in the analysis of adolescent health trends.

Figure 1 illustrates the year-by-year changes in mean BMI according to urban classification: Special City, Metropolitan City, and Province (Rural). All three groups showed a general increase in BMI up to 2022, followed by a gradual decline. Notably, students in provincial (rural) areas consistently exhibited the highest BMI across the observed years, and their rate of increase was also the most pronounced. This suggests that students in rural areas may have experienced a slower recovery in physical activity levels during the post-pandemic period.

Figure 2 visualizes trends in mean BMI by gender across the study years. Male students consistently demonstrated higher BMI values than female students across all time points, with the gap between the two groups widening considerably in 2022. This finding implies that when structural opportunities for physical activity—such as school-based programs—were restricted due to the pandemic, male students may have been more substantially affected in terms of reduced activity levels.

Figure 3 compares the mean BMI trends between capital and non-capital regions. Throughout the entire period, students in non-capital regions exhibited higher mean BMIs than their counterparts in the capital region, with the gap peaking between 2020 and 2022. This pattern offers structural implications, suggesting that spatial disparities in access to educational, physical activity, and public health resources were exacerbated during the pandemic, contributing to a widening gap in health indicators.

Particularly, the degree of change in BMI differed significantly by gender, with male students experiencing a more substantial increase in BMI during the pandemic years compared to female students. Moreover, regional differences—both between capital and non-capital areas and between urban and rural locations—showed distinct trajectories across years. These findings collectively support the interpretation that structural inequalities by region and gender may have played a significant role in shaping BMI outcomes during and after the pandemic, reinforcing the importance of a health equity perspective.

## 5. Discussion

This study analyzed longitudinal changes in BMI among Korean middle and high school students from 2018 to 2024, with a focus on differences before and after the COVID-19 pandemic across key intersecting variables such as sex, region, and urban–rural classification. The results showed a significant increase in BMI during the pandemic (2020–2022), peaking in 2022, followed by a gradual decline. Notably, male students and those in rural and non-capital regions consistently exhibited higher BMI levels, suggesting structural disparities in access to physical activity and health resources.

The average BMI levels observed in this study indicate that some groups exceeded the “normal” range as defined by PAPS classification standards. For instance, Figure 2 shows that the mean BMI of male students rose to approximately 23.4 in 2022, nearing the upper threshold of the normal range for second-year high school boys, with the potential for some subgroups to fall into the overweight category. In contrast, female students maintained an average BMI of around 21.7 during the same period, remaining within the normal range across all age groups. As shown in Figure 3, the average BMI of students in non-capital regions rose above 22.6 in 2022, approaching the upper limit of the normal range.

These year-to-year trends suggest that the COVID-19 pandemic imposed structural constraints on adolescents’ physical activity environments. The sharp rise in BMI between 2020 and 2022 reflects the impact of school closures, online learning, and restrictions on physical education and sports participation. These findings align with the results of [6], who reported declines in physical fitness and changes in body composition due to remote learning. Similarly, Ref. [19] demonstrated through a socio-ecological framework that routine behaviors such as commuting methods, sedentary time, and screen exposure can have long-term effects on BMI under environmental restrictions like pandemics.

The sex-specific analysis showed that male students consistently had higher BMI levels than female students, with a widening gap in 2022. This aligns with [16], who reported that although male students typically engage in more physical activity, they may experience a greater negative impact when activity opportunities are restricted. This finding calls for a re-examination of how physical education access and activity opportunities are distributed by sex.

Regional analysis revealed that students in non-capital and rural areas had higher BMI levels than those in capital and urban regions, with the disparity widening during the pandemic. This supports findings from [11,12], who highlighted the influence of regional differences in physical activity infrastructure, public health resources, and educational environments on adolescent health behaviors. Ref. [8], using the Socio-Ecological Model, also noted that physical activity among children and adolescents is more influenced by social and environmental conditions than by individual will. This study suggests that regional inequality has emerged as a form of dynamic inequality that expands over time.

These results underscore the need to interpret adolescent obesity not as an outcome of individual behavior alone but through the structural lenses of the SEM and HEF. Although the pandemic affected all students nationally, its impact varied significantly depending on sex, region, and social environment—largely due to differences in structural resource access and system responsiveness. Moreover, adolescent physical activity during the pandemic was shaped not only by external restrictions but also by internal psychosocial factors, calling for an integration of structural and qualitative approaches.

Ref. [20] explored barriers to and facilitators of physical activity from adolescents’ perspectives, identifying peer support, self-efficacy, enjoyment, and accessibility as key factors influencing engagement in moderate-to-vigorous physical activity (MVPA). This suggests that the effects of the pandemic were shaped by interactions between external restrictions and psychosocial factors such as motivation and attitude.

The HEF proposed by [9] posits that health disparities result from the interaction of multiple domains—including power structures, relational networks, biological susceptibility, and institutional arrangements—rather than from a single cause. This theoretical framework supports the need for structural and life-course approaches to interpret the observed disparities in BMI by sex and region.

Recent systematic reviews have also emphasized the effectiveness of multilevel socio-ecological interventions in reducing sedentary behavior and promoting physical activity [10]. This underscores the limitations of single-setting interventions and highlights the importance of coordinated strategies involving schools, families, communities, and policy. This perspective is also consistent with the equity-based intervention model discussed by [21], which emphasizes that addressing obesity requires not only education but also awareness of structural inequities and the redistribution of resources.

Furthermore, BMI increases during the pandemic cannot be attributed solely to reduced physical activity. Prior studies have shown that adolescents during this period experienced irregular eating patterns and increased consumption of snacks and high-calorie foods due to school closures and social isolation [7], as well as delayed sleep schedules and reduced sleep quality, which affected metabolic processes and appetite-regulating hormones [22].

Psychological factors must also be considered. The COVID-19 pandemic significantly increased levels of anxiety, depression, and stress among adolescents, which can lead to overeating, late-night eating, and emotional eating behaviors that directly or indirectly influence weight gain [23]. Thus, BMI changes during the pandemic should be understood as the result of a complex interplay between diet, sleep, emotional well-being, and physical activity. A one-dimensional interpretation risks overlooking this complexity.

The findings of this study highlight the importance of a multidimensional interpretation that incorporates a wide range of health determinants. Future research should adopt integrated analytical designs that include variables such as diet, sleep quality, emotional stability, and stress levels. This approach will provide a more holistic understanding of adolescent health and contribute to the development of effective interventions in future public health crises.

This study offers several strengths. First, it utilizes six years of large-scale national data, allowing for the analysis of longitudinal trends beyond short-term observations. Second, by considering multiple intersecting variables such as sex, region, and urban–rural classification, the study provides a comprehensive view of the structural determinants of adolescent health disparities. Third, the use of the SEM and HEF as theoretical frameworks enhances the explanatory power of the findings, allowing for practical policy recommendations beyond descriptive statistics.

Future school health and physical education policies must move beyond weight management and incorporate equity-based interventions that address intersectionality. Such policies are essential for protecting health rights during crises, reallocating resources across regions, and reducing sex-based disparities, thereby ensuring sustainable health equity for adolescents.

Nonetheless, this study has several limitations. The observed changes in BMI may partially reflect natural body composition changes associated with biological growth, which were not fully controlled for in the analysis. Adolescence is a period of rapid physical development, and age- or grade-based classifications may not fully capture individual differences in biological maturity. In addition, BMI is a relatively simple indicator compared to body fat percentage or detailed body composition metrics, and this study did not include mediating variables such as physical fitness or lifestyle behaviors. Future research should employ mixed-method designs that integrate both quantitative and qualitative data and adopt a life-course perspective in analyzing health inequalities.

## 6. Implications

This study offers the following theoretical implications. First, by applying a structural and socio-ecological analytical framework to explain changes in BMI, the study highlights the complexity and contextual nature of adolescent health behaviors. Second, through the application of the HEF, the study empirically demonstrates that the widening disparities in BMI by region and sex during the pandemic were not temporary outcomes, but rather the result of structural imbalances in access to resources. Third, by conducting a rare six-year longitudinal analysis using national data, this study provides a dynamic perspective on BMI trends and offers a meaningful evidence base for future public health policymaking in Korea.

## 7. Conclusions

This study utilized PAPS data from 2018 to 2024 to conduct a time-series analysis of changes in BMI among Korean adolescents before and after the COVID-19 pandemic. The results revealed a significant increase in average BMI during the pandemic period, with particularly elevated levels observed among male students and those residing in non-capital and rural areas. Although a gradual decline in BMI was observed in the post-pandemic years, clear disparities by sex and region persisted. These findings provide empirical evidence that changes in adolescent health indicators are not merely the result of individual-level factors, but are closely linked to structural issues such as educational environments, regional public health infrastructure, and differential access to resources.

## Figures and Tables

**Figure 1 ijerph-22-01136-f001:**
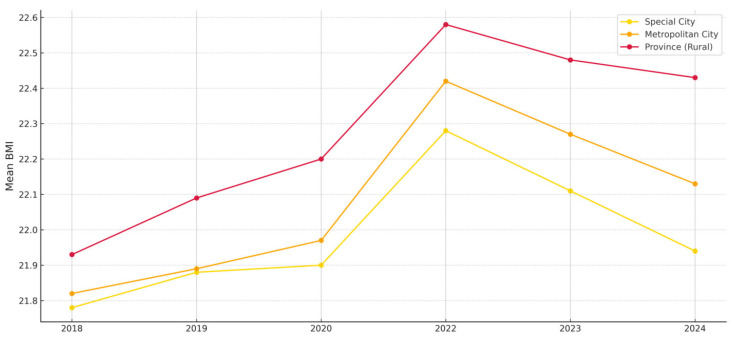
Changes in Mean BMl by Urban-Rural Classification (2018–2024).

**Figure 2 ijerph-22-01136-f002:**
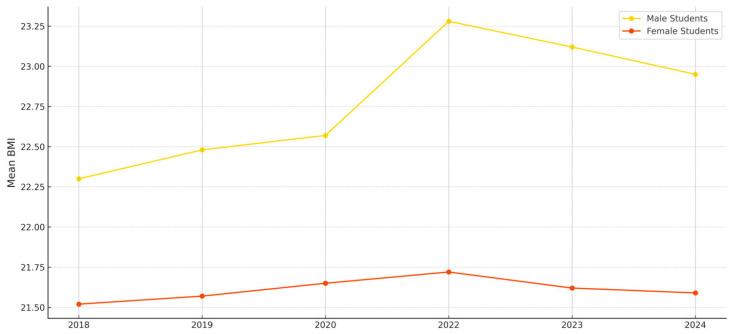
Changes in Mean BMl by Gender (2018–2024).

**Figure 3 ijerph-22-01136-f003:**
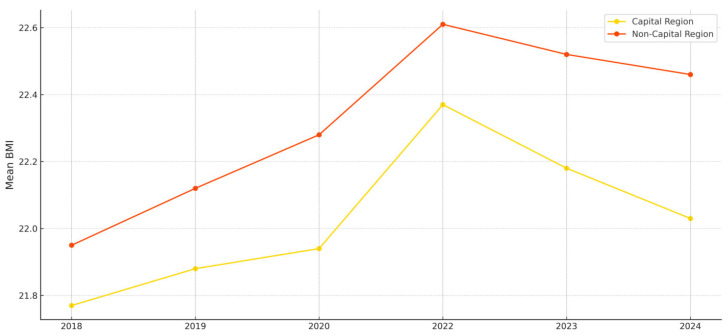
Changes in Mean BMl by Region (2018–2024).

**Table 1 ijerph-22-01136-t001:** Descriptive statistics of participants.

Category		Frequency (n)	Percentage (%)
Year	2018	28,322	16.5
	2019	28,314	16.5
	2020	28,414	16.5
	2022	28,710	16.7
	2023	28,912	16.8
	2024	29,033	16.9
Region	Capital Region	66,186	38.5
	Non-Capital Region	105,519	61.5
Gender	Male	86,235	50.3
	Female	85,328	49.7
Administrative Area	Special City	20,836	12.1
	Metropolitan City	37,200	21.7
	Province	113,669	66.2

**Table 2 ijerph-22-01136-t002:** BMI classification standards in PAPS.

Grade Level	Underweight		Normal		Overweight		Mild Obesity		Severe Obesity	
	Male	Female	Male	Female	Male	Female	Male	Female	Male	Female
Middle 1st	≤15.4	≤15.5	15.5– 22.6	15.6– 22.9	22.7– 24.7	23.0– 25.7	24.8– 34.7	25.8– 35.8	≥34.8	≥35.9
Middle 2nd	≤16.1	≤15.9	16.2– 23.1	16.0– 23.3	23.2– 25.6	23.4– 26.2	25.7– 35.6	26.3– 36.3	≥35.7	≥36.4
Middle 3rd	≤16.7	≤16.5	16.8– 23.7	16.6– 23.7	23.8– 25.9	23.8– 26.6	26.0– 35.9	26.7– 36.7	≥36.0	≥36.8
High 1st	≤17.3	≤16.9	17.4– 23.8	17.0– 24.1	23.9– 26.0	24.2– 26.8	26.1– 36.0	26.9– 36.9	≥36.1	≥37.0
High 2nd	≤17.6	≤17.4	17.7– 23.7	17.5– 24.3	23.8– 25.9	24.4– 26.9	26.0– 35.9	27.0– 37.0	≥36.0	≥37.1
High 3rd	≤18.0	≤17.9	18.1– 23.8	18.0– 24.5	23.8– 25.5	24.6– 26.8	25.6– 35.5	26.9– 36.9	≥35.6	≥37.0

**Table 3 ijerph-22-01136-t003:** One-way ANOVA results for differences in mean BMI by year and urban–rural classification.

Source of Variation	Sum of Squares	df	Mean Square	F
Year	7843.266	5	1568.653	385.173 ***
Urban–Rural Classification	2418.638	2	1209.319	294.642 ***

*** *p* < 0.001

**Table 4 ijerph-22-01136-t004:** Comparison of mean BMI by gender and region.

Variable	Group	M	SD	t
Gender	Male	22.775	1.914	121.36 ***
	Female	21.63	1.979	
Region	Capital Region	22.016	1.467	−30.614 ***
	Non-Capital Region	22.325	2.306	

*** *p* < 0.001

**Table 5 ijerph-22-01136-t005:** Two-way ANOVA results for interaction effects on mean BMI.

Source of Variation	Sum of Squares	df	Mean Square	F
Year	4146.517	5	829.303	204.409 ***
Urban–Rural Classification	2410.785	2	1205.393	297.109 ***
Year × Urban–Rural Classification	271.318	10	27.132	6.687 ***
Error	690,931.287	170,303	4.057	
Total	701,467.232	170,320		
Year	7773.269	5	1554.654	417.906 ***
Gender	55,589.9	1	55,589.9	14,943.109 ***
Year × Gender	4235.104	5	847.021	227.687 ***
Error	633,566.969	170,309	3.72	
Total	701,467.232	170,320		
Year	7146.071	5	1429.214	353.095 ***
Region	3852.510	1	3852.51	951.783 ***
Year × Region	388.420	5	77.684	19.192 ***
Error	689,355.369	170,309	4.048	
Total	701,467.232	170,320		

*** *p* < 0.001

## Data Availability

The data presented in this study are openly available at the National Education Information System (NEIS) website: https://www.neis.go.kr (accessed on 26 April 2025).

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
