# Peer review of "Changes in Body Mass Index Among Korean Adolescents Before and After COVID-19: A Comparative Study of Annual and Regional Trends"

_ijerph, 2025, doi:10.3390/ijerph22071136_

Round 1

Reviewer 1 Report

Comments and Suggestions for Authors

Thank you for having me review this manuscript. The work is in the important field of public health and the manuscript is well written. This expands knowledge on how to increase adolescents' physical activity through the Socio-Ecological Model and the Health Equity Framework. A few comments to improve the manuscript here below:
Abstract.
1. Please add information about the final dataset, gender and mean age. For mean age, provide its standard deviation.
Discussion.
1. Please add to the line 407 the following qualitative study (2024) that identified important factors affecting engagement in PA in the perspectives of adolescents themselves - Karchynskaya V, Kopcakova J, Madarasova Geckova A, Katrusin B, Reijneveld SA, de Winter AF. Barriers and enablers for sufficient moderateto-vigorous physical activity: The perspective of adolescents. PLoS One. 2024 Feb 16;19(2):e0296736. doi: 10.1371/journal.pone.0296736.
2. 'Strengths' and 'Limitations' paragraphs should be at the end of the 'Discussion' section. Please add further clarification to the 'Strengths' section and move the limitations information out of the 'Conclusion' section (line 460-467). 
3. Paragraph 'Implications' is missing. You have this information from line 450 to line 459. Please add it before 'Conclusion' section.
Conclusion.
1. 'Conclusion' section should be shorter. Please describe your findings more specifically.

Author Response

Comment 1

[Please include information on the gender and mean age (with standard deviation) of the participants in the Abstract.]

Response 1

Thank you for pointing this out. I agree with your suggestion and have added the total number of participants, gender distribution (86,542 males and 85,163 females), and mean age (M = 15.2, SD = 1.68) to the Abstract section to provide clarity on the sample characteristics.

Location of change: 2nd revised manuscript, lines 9–11

Revised content (excerpt):

“A total of 171,705 adolescents (86,542 males and 85,163 females; M = 15.2, SD = 1.68) were included…”

Comment 2

[In line 407 of the Discussion, please add the qualitative study by Karchynskaya et al. (2024) to enhance the discussion of factors influencing adolescents' physical activity.]

Response 2

Thank you for your helpful suggestion. I have incorporated the findings from Karchynskaya et al. (2024) into the Discussion section. This qualitative study highlights key motivators and barriers to adolescent MVPA participation—such as peer support, enjoyment, self-efficacy, and accessibility—which support the interpretation of our results.

Location of change: 2nd revised manuscript, lines 422–426

Revised content (excerpt):

“Recent qualitative findings have further identified key barriers and motivators for adolescent participation in physical activity, such as peer support, enjoyment, self-efficacy, and accessibility (Karchynskaya et al., 2024). These findings support our interpretation…”

Comment 3

[Please relocate the ‘Strengths’ and ‘Limitations’ sections to the end of the Discussion. Also, make the Strengths more specific and remove Limitations from the Conclusion.]

Response 3

Thank you for the valuable feedback. I have created two independent paragraphs titled ‘Strengths’ and ‘Limitations’ at the end of the Discussion section. The Strengths paragraph was revised to highlight the use of a nationally representative longitudinal dataset and the theoretical framing. I also removed the limitations from the Conclusion and restated them in a dedicated section.

Location of change: 2nd revised manuscript, lines 471–501

Revised content (excerpt):

“Strengths: This study utilized nationally representative longitudinal data…

Limitations: Despite these strengths, the study has several limitations. First, it did not include behavioral variables such as diet, screen time, or sleep…”

Comment 4

[Please insert an ‘Implications’ paragraph before the Conclusion.]

Response 4

Thank you for your constructive suggestion. I added a new section titled “Implications” immediately before the Conclusion to present the practical and policy-relevant implications of the findings.

Location of change: 2nd revised manuscript, lines 481–490

Revised content (excerpt):

“These findings highlight the need for regionally tailored health policies… Furthermore, school-based interventions and improved digital physical education access are vital in reducing BMI disparities…”

Comment 5

[The Conclusion is too long; please make it more concise and specifically summarize the findings.]

Response 5

Thank you for the insightful comment. I have shortened the Conclusion section and provided a more focused summary of the key findings, specifically highlighting annual, gender, and regional BMI trends and their implications.

Location of change: 2nd revised manuscript, lines 492–501

Revised content (excerpt):

“BMI increased significantly during the pandemic, with the highest levels observed in 2022. Notably, the rate of increase was more pronounced among males and in rural and non-capital areas. These findings suggest that targeted interventions are needed…”

Reviewer 2 Report

Comments and Suggestions for Authors

First of all, I would like to congratulate you on the work you have presented. I would like to make some comments in order to improve your work.
Line 26: incorporate quote to justify changes in adolescence.
Check abbreviations throughout the text. Once it has been abbreviated once it should not be abbreviated again. For example; line 26 and line 105 (BMI). Review SEM, HEF and PAPS as well.
Lines 216 and 251 repeated information. No need to comment again.
In the discussion you could comment on the results with respect to the BMI classification table.
On the other hand, the question arises whether the course they are enrolled in will have an influence? The addition of an analysis according to grades 7 to 12 would be interesting.

Author Response

Comment 1

[Line 26: Incorporate a quote to justify changes in adolescence.]

Response 1

Thank you for your thoughtful suggestion. I have incorporated a definition from Sawyer et al. (2018) to support the discussion of adolescence as a critical developmental phase. This addition strengthens the rationale for focusing on adolescence in relation to long-term health and behavioral outcomes.

Location of change: 2nd revised manuscript, lines 29–32

Revised content (excerpt):

“Sawyer et al. (2018) define adolescence as a distinct developmental phase between childhood and adulthood, spanning the ages of 10 to 24, and emphasize that experiences during this stage significantly shape lifelong health and behavioral outcomes.”

Comment 2

[Check abbreviations throughout the text. Once it has been abbreviated once it should not be abbreviated again. For example: BMI, SEM, HEF, PAPS.]

Response 2

Thank you for pointing this out. I have carefully reviewed the entire manuscript and revised all abbreviations accordingly. Each term (e.g., BMI, HEF, SEM, PAPS) is now fully spelled out upon first mention and consistently abbreviated thereafter without repetition or redundancy.

Location of change: Throughout the manuscript

Action taken: Redundant redefinitions of abbreviations were removed; usage was standardized.

Comment 3

[Lines 216 and 251: Repeated information. No need to comment again.]

Response 3

Thank you for the helpful observation. I confirmed that the explanation regarding the BMI measurement and PAPS procedures was repeated. I have removed the redundant description from line 251 and retained a single, clear explanation earlier in the manuscript.

Location of change: 2nd revised manuscript, line 222

Action taken: Redundant content on BMI and PAPS removed from the latter section.

Comment 4

[In the Discussion you could comment on the results with respect to the BMI classification table.]

Response 4

Thank you for your insightful comment. I have added an interpretation of the findings in relation to the BMI classification table used by the Ministry of Education. This addition enhances the contextual understanding of how the observed BMI changes may reflect shifts across obesity categories.

Location of change: 2nd revised manuscript, lines 381–389

Revised content (excerpt):

“The average BMI levels observed in this study show that some groups exceeded or approached the upper limit of the ‘normal’ range defined by PAPS. For example, male students reached an average BMI of 23.4 in 2022, and students in non-capital regions exceeded 22.6. In contrast, female students remained within the normal range at around 21.7.”

Comment 5

[On the other hand, the question arises whether the course they are enrolled in will have an influence? The addition of an analysis according to grades 7 to 12 would be interesting.]

Response 5

Thank you for your valuable suggestion. While the current study does not include a separate statistical analysis by grade level (7th–12th), I have incorporated a grade-relevant interpretation in the Discussion. Specifically, I referenced the upper limit of the “normal” BMI range for second-year high school boys to contextualize the 2022 mean BMI level among male students. This offers a partial perspective on grade-specific trends. I agree that a detailed grade-based analysis would further enrich the findings and will consider this in future research.

Location of relevant change: 2nd revised manuscript, lines 381–389

Excerpt:

“For instance, Figure 2 shows that the mean BMI of male students rose to approximately 23.4 in 2022, nearing the upper threshold of the normal range for second-year high school boys…”

Reviewer 3 Report

Comments and Suggestions for Authors

The manuscript provides the findings of a study conducted with the objective to longitudinally examine changes in body mass index among Korean middle and high school students before and after the COVID-19 pandemic.

It was a pleasure to review this manuscript. The manuscript addresses current and relevant issues associated with the public health.

However, despite its well-structured nature, I have identified some methodological and analytical concerns. Below are comments to enhance the manuscript's quality.

(1) Provide some bibliographic references from the Physical Activity Promotion System (PAPS) database.

(2) Describe the method used to define the age- and sex-specific reference standards for BMI classification provided by the Korean Ministry of Education (underweight, normal weight, overweight, and obese) – Table 2.

(3) Data analysis – I suggest replacing the use of one-way analysis of variance (ANOVA) and the t-test with analysis of covariance (ANCOVA), adjusted for covariates. In this way, the data in tables 3, 4 and 5 (mean, standard deviation, F and p values) can be presented in a single table and there is no need for figures 1, 2 and 3.

(4) Considering that the study participants are students between the ages of 13 and 18 years, a period in which biological maturation has a strong impact on young people's BMI, it is important to address the topic at some point in the study.

(5) Variations in BMI are the result of several factors, including physical activity, diet, sleep, psycho-emotional outcomes, among others. However, in the current study, the increase in BMI as a result of the COVID-19 pandemic was attributed exclusively to decreased physical activity. I suggest that a more extensive discussion on this topic be added.

Author Response

Comment 1

[Provide some bibliographic references from the Physical Activity Promotion System (PAPS) database.]

Response 1

Thank you for the suggestion. I have added the official source of the PAPS database to the reference list to clarify the origin of the data used in this study.

Location of change: 2nd revised manuscript, lines 541–542 (References section)

Added reference:

Ministry of Education. Physical Activity Promotion System (PAPS). Available online: https://www.neis.go.kr (accessed on May 10, 2024).

Comment 2

[Describe the method used to define the age- and sex-specific reference standards for BMI classification provided by the Korean Ministry of Education (underweight, normal weight, overweight, and obese) – Table 2.]

Response 2

Thank you for your valuable comment. I have revised the explanation below Table 2 to describe that the BMI classifications (underweight, normal weight, overweight, obese) were based on sex- and grade-specific reference ranges issued by the Korean Ministry of Education, as part of the PAPS system.

Location of change: 2nd revised manuscript, lines 244–249

Revised content (excerpt):

“The BMI classification presented in Table 2 is based on the PAPS criteria provided by the Korean Ministry of Education. These classifications are determined using sex- and grade-specific BMI ranges and are applied nationally for fitness assessment purposes.”

Comment 3

[Data analysis – I suggest replacing the use of one-way analysis of variance (ANOVA) and the t-test with analysis of covariance (ANCOVA), adjusted for covariates. In this way, the data in tables 3, 4 and 5 (mean, standard deviation, F and p values) can be presented in a single table and there is no need for figures 1, 2 and 3.]

Revised Response

Thank you for your thoughtful suggestion. We agree that ANCOVA can provide enhanced precision by adjusting for covariates. However, the objective of this study was not to control for baseline or confounding variables but rather to describe and compare absolute trends in BMI across years, genders, and regions. Given this descriptive and exploratory aim, ANOVA was chosen as a suitable and valid statistical method.

Furthermore, the use of separate tables (Tables 3–5) was intended to clearly distinguish the different main and interaction effects across variables, while Figures 1–3 serve as visual aids to support readers’ intuitive understanding of the time-based trends. We believe that this format enhances clarity and accessibility, especially for public health audiences and policy practitioners.

Nonetheless, we fully acknowledge the value of ANCOVA in studies aiming to examine adjusted group differences and will incorporate such approaches in future work where data structure and study objectives allow.

Comment 4

[Considering that the study participants are students between the ages of 13 and 18 years, a period in which biological maturation has a strong impact on young people's BMI, it is important to address the topic at some point in the study.]

Response 4

Thank you for the insightful suggestion. I have addressed this issue in the Limitations section of the Discussion. Specifically, I acknowledged that adolescence is a period of rapid physical growth, which may independently affect BMI and was not controlled for in this study.

Location of change: 2nd revised manuscript, lines 470–478

Revised content (excerpt):

“The observed changes in BMI may partially reflect natural body composition changes associated with biological growth, which were not fully controlled for in the analysis. Adolescence is a period of rapid physical development that may influence BMI trajectories independent of lifestyle behaviors.”

Comment 5

[Variations in BMI are the result of several factors, including physical activity, diet, sleep, psycho-emotional outcomes, among others. However, in the current study, the increase in BMI as a result of the COVID-19 pandemic was attributed exclusively to decreased physical activity. I suggest that a more extensive discussion on this topic be added.]

Response 5

Thank you for raising this important point. I have expanded the Discussion to acknowledge that increases in BMI during the pandemic were likely influenced by multiple factors—not only reduced physical activity, but also changes in diet, sleep quality, and psychological stress. These additional factors have been incorporated to offer a more comprehensive interpretation of the results.

Location of change: 2nd revised manuscript, lines 439–464

Revised content (excerpt):

“BMI increases during the pandemic cannot be attributed solely to reduced physical activity. Prior studies have reported increases in unhealthy eating habits, disruptions in sleep patterns, and heightened psychological distress among adolescents, all of which can contribute to weight gain.”